# GENERATING ROBOT POLICY CODE FOR HIGH-PRECISION AND CONTACT-RICH MANIPULATION TASKS

## ABSTRACT

Large Language Models (LLMs) have been successful at generating robot policy code, but so far these results have been limited to high-level tasks that do not require accurate movement. It is an open question how well such approaches can work for high-precision, contact-rich tasks that require controlling contact forces with the environment. We find that, with the right action space, LLMs are capable of successfully generating policies for a variety of contact-rich and high-precision manipulation tasks in a zero-shot fashion. Specifically, we reparameterize the action space to include robot compliance with constraints on the interaction forces and stiffnesses involved in reaching a target pose. We validate this approach on subtasks derived from the Functional Manipulation Benchmark (FMB) and the IROS 2020 Robotic Grasping and Manipulation Competition, where zero-shot policy generation in this action space improves success rates over non-compliant action spaces by greater than 3x and 4x, respectively, over a baseline that uses free space motions. To further investigate properties that make language models well posed to generate contact-rich tasks, we also analyse language models ability to complete control-relevant arithmetic reasoning tasks over continuous numbers in-context and ablate the importance of different prompt components in generating relevant motion patterns. Project webpage: https://dex-code-gen.github.io/dex-code-gen/

## 1 INTRODUCTION

Many of the open problems in learning-based robotics revolve around the issue of scaling: deep-learning methods require vast datasets that are not readily available for robotics applications. One workaround for the data scarcity problem is to retrofit large models that have been trained on internet-scale datasets from other modalities for robotics tasks. Recently, large language models (LLMs) have emerged as a strong candidate for this approach. LLMs are able to successfully generate programming code, complete numeric sequences, and solve common-sense reasoning tasks (Liang et al., 2022; Mirchandani et al., 2023; Huang et al., 2022a). Because code is one of the most popular interfaces for specifying robotic planning and control commands, these capabilities hint at enormous potential when applied to robotics.

Past work demonstrated that generating robot policy code from LLMs is successful for high-level tasks such as navigation and open-vocabulary pick-and-place (Liang et al., 2022; Mirchandani et al., 2023). For example, a language model can compose high-level action primitives such as `grab(chips)` or `move_to(human)` to generate a successful policy conditioned on a natural language command like "bring me the chips" (Liang et al., 2022). But at present, lower-level tasks and behaviors are generally considered out of reach for LLMs and to the best of our knowledge, there have not been any compelling demonstrations of such capabilities with these tools.

While various robot learning approaches have been able to demonstrate impressive generalization across different settings and target objects for pick-and-place tasks (Jiang et al., 2022; Shridhar et al., 2022; Brohan et al., 2022; 2023; Shridhar et al., 2023), such generalization is arguably more difficult for dexterous tasks where a higher level of precision is required. For example, for a peg-in-hole insertion task, surfaces with more friction or tight insertion tolerances may require multiple insertion attempts or contact force tuning to reach the insertion site. Similarly, pegs with different geometries may need different approach trajectories to achieve proper alignment: a peg with a star-shaped cross-section may require an initial rotation for insertion, whereas no such rotation is required

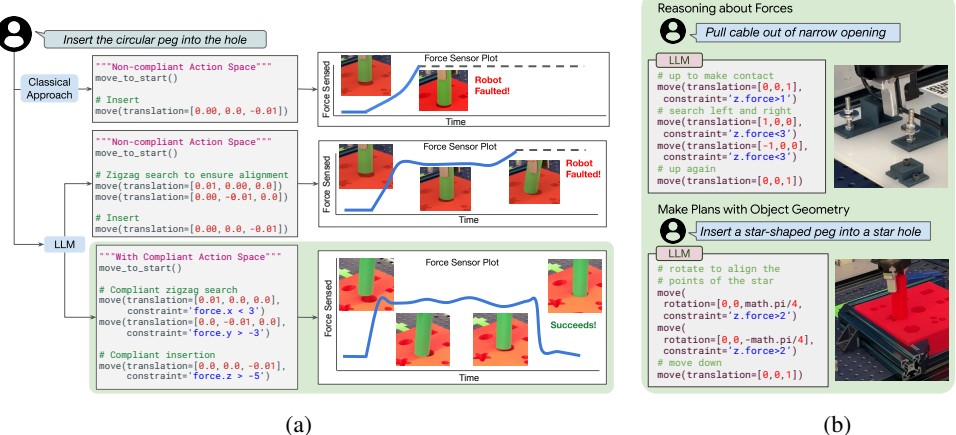

(a)                                      (b)

Figure 1: (a) We prompt an LLM to generate code for high-precision tasks. By using an action space that includes parameters that enable compliant behavior, the LLM is able to generate action sequences that successfully complete contact-rich manipulation tasks like this peg insertion. (b) Language models' ability to reason about object geometry and make plans by using world knowledge about different object types enables zero-shot generalization to new objects and scenes.

for a peg with a circular cross-section. In practice, the parameters of contact-rich insertion skills are mostly tuned by experts to handle these differences and automating this process is still an open problem. Therefore, directly providing a language model with a library of more dexterous skills (such as `insert(peg)`) will not work out of the box.

In this paper, we propose a promising alternative for automating the parameter-tuning process within the control API, where we aim to leverage the world knowledge inside language models to set control parameters. In particular, our goal is to understand if LLMs have the ability to reason about motions and forces acting on objects, such as to generalize over a much larger class of objects and robot manipulations. To study this topic, we modify the action space in which a language model operates by exposing constraints on the contact stiffness and forces observed in the process of contact-rich manipulation. These modification open up the possibility to study code generation for contact-rich tasks, including industry relevant tasks such as high precision insertion, rigid body assembly, and deformable object manipulation (see Figure 1).

The main contribution of this work is to demonstrate that LLMs, without any specialized training, have the ability to perform contact-rich tasks when given the appropriate action space. In particular, we develop a system for automatically generating robot policy code for dexterous tasks by allowing LLMs to specify constraints on the stiffness, forces, and trajectories required to perform contact-rich manipulation tasks. We show that this approach is able to outperform a contact-unaware model by over 3x on average on subtasks developed from two challenging contact-rich benchmarks. Specifically, this approach is able to generate novel insertion patterns from high level descriptions of object shape and texture on insertion tasks from the Functional Manipulation Benchmark (FMB) (Luo et al., 2023) and to route and un-route cables in the style of the IROS 2020 Robotic Grasping and Manipulation Competition (IROS RGMCS) (Sun et al., 2021). We hope that our demonstration could provide a step towards adapting language models to generate robot code for more dexterous tasks and unlock the benefits of internet-scale data and foundation models for robotics applications.

## 2 RELATED WORK

**LLMs in robotics.** Past work demonstrated that LLMs can successfully generate robot policy code for pick-and-place style manipulation tasks (Huang et al., 2022b), compose mid-level plans for navigation tasks (Huang et al., 2022a), and compose multiple navigation and manipulation skills for integrated household agents (Singh et al., 2023; Wu et al., 2023). Many of these approaches rely on filtering LLM-generated code based on what is executable (Gai et al., 2021) or on making hierarchically queries (Liang et al., 2022). However, to the best of our knowledge, it is not yet established in the literature whether LLMs can generate robot policy code for performing high-precision contact-rich manipulation tasks, which we study in this work.

**Arithmetic Reasoning and Pattern Extrapolation Abilities of LLMs.** Mirchandani et al. (2023) demonstrated that models trained on internet-scale data are capable of doing general spatial and sequential reasoning tasks. Similar to our work, their analysis is inspired by robot control-relevant reasoning problems. However, unlike them, we produce a proof of concept on real hardware and on a challenging contact rich manipulation task. Our arithmetic reasoning tasks are inspired by work from Garg et al. (2022), which shows that ~~the~~ Transformers (Vaswani et al., 2017) can learn simple function classes in-context after training on regression problems. Unlike them, we show that this capability emerges in models trained only on text generation.

**Contact-rich robot manipulation** tasks are those that involve a robot making controlled contact with its environment while performing them. These tasks constitute a vast majority of manipulation tasks in daily life, including household tasks such as wiping tables and sweeping dust into a dust-pan (Wi et al., 2023), and industrial tasks such as high precision insertion (Luo et al., 2019; Zhao et al., 2022) and rigid body assembly (Narang et al., 2022; Liu et al., 2022). A robot needs to reason about the contact forces it will impart and experience from the environment while performing such tasks to complete them successfully. Learning a general policy to perform a wide array of high-precision contact-rich manipulation tasks has been studied in great detail in the robotics literature (Kroemer et al., 2021), (Suomalainen et al., 2022; Elguea-Aguinaco et al., 2023; Zhao et al., 2022; Morgan et al., 2021; Davchev et al., 2022; Luo et al., 2019; Migimatsu et al., 2022), yet how to find a general approach to these tasks remains an open question. Prior work directly learns policies with imitation learning Chi et al. (2023) or reinforcement learning Schoettler et al. (2019); Narang et al. (2022); Brahmbhatt et al. (2023), but these require hundreds of human demonstrations, significant operator training, dedicated simulators, or thousands of environment interactions to achieve a performant policy. In this work, we step towards obtaining a general policy for high-precision contact-rich manipulation tasks by leveraging the world knowledge inside LLMs and combining it with the appropriate task action spaces. We choose robot impedance (or equivalently, admittance) control as the action space for contact-rich robot manipulation tasks as it can regulate the relationship between robot position and contact forces effectively (Beltran-Hernandez et al., 2020; Abu-Dakka & Saveriano, 2020).

## 3 PRELIMINARIES

Our goal is to develop a system that can translate natural language (NL) instructions into robotic actions by leveraging a sufficiently expressive API for control. Past work (Liang et al., 2022) has shown that off-the-shelf language models can be adapted towards this goal with few-shot prompting. Concretely, pairs of natural language requests with corresponding robot policy code

```
# You're a robot trying to insert a
   peg in a hole. Grab the circular peg.
pick_up(circular_peg)
```

Figure 2: LLM-generated code (highlighted in blue) from a NL request.

are fed into a language model. Then, the language model can output novel programs in response to new commands as shown in Figure 2. The success of this approach can be attributed to the fact that during offline training on vast internet datasets language models absorb world knowledge about common-sense interactions and learn mappings between natural language instructions and code. Strategies for adapting this approach towards a contact-rich setting are discussed in the next section.

## 4 GENERATING CONTACT-RICH POLICY CODE WITH LANGUAGE MODELS

We begin by describing the prompting strategies for contact-rich high-precision manipulation tasks. Equipped with a set of prompts, we then discuss the different choices of robotic action spaces that can be made available to a language model including our proposed action space.

### 4.1 PROMPTING FOR CONTACT-RICH CONTROL

We consider five prompting strategies when generating robot policy code from a language model:

1. **Task descriptions** are high-level descriptions of the scene and the task goal written in natural language. These can occur at both the beginning and end of a prompt and often include important information about the task setup such as the peg shape or the available objects. See Fig. 3a for an example.

2. **Descriptions of available control APIs** are formatted doc-strings that describe the code that can be used by the LLM. These include lists of available variables as well as the expected range of values for floating point numbers. Fig. 3b shows an example description for the available `move`

function. We also include descriptions for the full library of available methods, which includes a point-to-point move, a compliant move, conditions, gripper movements, and methods or variables specifying the positions of relevant objects.

3. **Hints** in our setting include rules, keywords that specify relevant control primitives, and requests to have the model explain its reasoning in natural language or in pseudocode. Phrases such as "perform a pattern search" guide the model towards predicting behavior that better recovers from errors and better handles imprecision in the position of target poses. Intuitively these keywords help reducing task ambiguity (e.g., by emphasizing that provided locations are imprecise) and guide the model towards motion patterns that are relevant to contact-rich tasks. Requests to explain in natural language can be thought of as a variant of chain-of-thought prompting Kojima et al. (2022). The specific keywords and requests that are helpful in each task are described in the experimental section.

4. **Spatial patterns** are symbolic summaries of a given scene and act as a character-based representation of what is visible to the agent. An example of a symbolic representation of an IROS RGMCS (Sun et al., 2021) board is summarized in Figure 3d. The `c` symbol refers to the path of the cable, `S` refers to a screw, and `B` refers to a plastic channel component through which the cable is routed. These would also be defined in the prompt. Although these are specified by the prompt designer in our examples, they could also be generated by a vision-language model or other perception APIs (Wi et al., 2023).

5. **Examples** of the control APIs being used for basic movements, such as making contact with a surface, are useful for tasks with ambiguity or where the desired force constraints are difficult to infer from the given ranges.

```
"""You're a robot trying to undo
  cable routing. Unroute the cable
  from the screws and brackets it is
  wrapped around."""
```

(a) Task Description

```
"""Use these methods:
- move: moves to specified offset
  Args:
      translation: (x, y, z) tuple
      rotation: (x, y, z) tuple
..."""
```

(b) Control API Descriptions

```
"""Because the bracket opening is
  small, the cable is very prone to
  getting caught in the opening."""
```

(c) Hints

```
"""Here is a board layout:
    c c c     c c c
    c S c     B   c
    c   c S c     B   c
    c   c c c     c c c"""
```

(d) Spatial Patterns

```
# Move the cable to the left until
  it snags
move((1, 0, 0),
    constraint=(x.force>-1))
```

(e) Examples

Figure 3: Categories of prompts.

The combination of the prompt strategies described above allows us to prompt a language model with enough contextual information about the dexterous task at hand. Next, we discuss how we can design the action space of the robot to be able to perform such tasks in practice.

## 4.2 ACTION SPACES FOR ROBOT MANIPULATION TASKS

Past approaches assume access to a library of methods that exhaustively cover all user-requested commands (Liang et al., 2022; Mirchandani et al., 2023). Building such a library is challenging for contact-rich tasks because in practice these policies are tuned by experts across different object geometries, frictions, and scene layouts. This section describes different approaches to parameterizing the control API and what the right choice of a control API can achieve. Formally, we consider a contact-rich robot manipulation task to be composed of a sequence of subtasks $\tau = (t_1, \ldots, t_n)$. The specific definition of a subtask will change based on the action space, as described below.

**Point-to-point moves.** In the past work, (Liang et al., 2022; Mirchandani et al., 2023), the authors make use of an action space that directly command the robot to `move` to target poses in the Cartesian space, $[\mathbf{x}_{target}]_i$ (See Fig. 4). In this setting, each sub-task $t_i$ is simply defined as the next Cartesian pose (a.k.a. waypoint): $t_i = ([\mathbf{x}_{target}]_i)$.

```
# Insert a peg into a hole
pick_up(peg)
# go down to make contact
move([0, 0, -1])
# wiggle to find opening
move([1, 0, 0])
move([-1, 0, 0])
# go down to insert
move([0, 0, -1])
```

Figure 4: Generated code with free space motions

While this approach is successful for executing motions in free-space or for simple pick-and-place tasks, it fails when the robot needs to explicitly make a purposeful contact with its environment. Consider a robot trying to make contact with a surface to perform a wiping motion. Successfully parameterizing a

policy in this action space would require predicting a precise Cartesian pose with very little tolerance for error. Predicting millimeters short of the surface would fail to make a contact and predicting millimeters too deep into the surface would cause the robot to exert high forces on the surface, which can cause faults in the robot or even break it in the worst case scenario.

**Compliant moves.** Addressing this shortcoming, we propose to parameterize the action space for performing contact-rich manipulation tasks using robot's compliance, realized in impedance control (or equivalently admittance control which has been shown to be an adequate action space for robot learning in Martín-Martín et al. (2019)). An impedance move action is parameterized by both a target Cartesian pose, $[\mathbf{x}_{target}]_i$, and a vector that specifies stiffness along each degree of freedom, $\sigma_i$: $t_i = ([\mathbf{x}_{target}]_i, \sigma_i)$, when the robot is in contact with the environment. During execution, the stiffness vector for each subtask can be used to define the parameters for a variable impedance controller (Buchli et al., 2011) of the form:

$$F_{external} = K_p(\mathbf{x}_d - \mathbf{x}) + K_d(\dot{\mathbf{x}}_d - \dot{\mathbf{x}}) + \Lambda(\ddot{\mathbf{x}}_d - \ddot{\mathbf{x}}) \tag{1}$$

where $\mathbf{x}_d$, $\mathbf{x}$, $\dot{\mathbf{x}}_d$, $\dot{\mathbf{x}}$, $\ddot{\mathbf{x}}_d$, and $\ddot{\mathbf{x}}$ denote the target and current Cartesian pose, twist, and accelerations, respectively. $K_p$, $K_d$, and $\Lambda$ correspond to the stiffness, the damping, and the task-space inertia matrices, respectively. The impedance controller realizes that the robot's end-effector in contact with the environment behaves like the linear spring-damper-mass system above. $K_p$, $K_d$, and $\Lambda$ are computed as a function of our specified stiffness vector $\sigma_i$ (explained below) and robot specific parameters in order to achieve stable yet responsive robot behavior (see Appendix).

Intuitively, the stiffness vector determines the interaction forces that the robot will impart on its environment while performing the task. Low stiffness coefficients in $\sigma$ regulate the robot's compromise between contact forces and the attempt to achieve position accuracy. In the example that we discussed in the last paragraph, a low stiffness value would enable the robot to maintain gentle contact with a surface that prevents the robot from reaching a desired position. A higher stiffness value would create higher contact forces, equivalent to a higher priority to reduce position error.

```
# Insert a peg into a hole
pick_up(peg)
# go down to make contact
move([0, 0, -1],
    constraint=(z.force>1))
# wiggle to find opening
# stop when force lessens
move([1, 0, 0],
    constraint=(z.force<1))
move([-1, 0, 0],
    constraint=(z.force<1))
# go down to insert
move([0, 0, -1],
    constraint=(z.force>2))
```

Figure 5: Generated code with impedance moves and constraints.

**Conditional compliant moves.** In addition to the impedance control specification described above, we also allow the LLM to specify the set of conditions under which to terminate an impedance move. Specifically, these are thresholds on force or position in a specified coordinate direction. An example pseudocode is presented in Fig. 5. This is a powerful primitive as it enables the robot to construct recipes for high-precision tasks that do not rely on fine-grained visual perception. In the example of making contact with a surface, this may look like moving a peg downwards with a termination constraint on upward force.

## 5 EXPERIMENTS: HIGH-PRECISION CONTACT-RICH MANIPULATION TASKS

In this section, we evaluate the ability of LLMs to generate code for fine-grained manipulation tasks that require high precision in a series of experiments. First, we perform a set of isolated experiments in order to establish the general feasibility of this approach to arithmetic reasoning tasks, which are a prerequisite to the numerical reasoning needed for effectively generating robotic controller code (Section 5.1). Second, we move on to evaluating our proposed approach on a set of real robotic tasks, namely, a subset of high-precision contact-rich manipulation tasks from the Functional Manipulation Benchmark (Luo et al., 2023) and a set of industrial manipulation tasks adapted from the IROS 2020 Robotic Grasping and Manipulation Competition (IROS RGMCS) (Sun et al., 2021) (Section 5.2). Later, we evaluate prompt hint ablations to study the utility of incorporating additional hints in generating relevant motion patterns for robot manipulation tasks (Section 5.3).

### 5.1 REASONING OVER CONTINUOUS SPACES

While prior work has demonstrated that LLMs are capable of serving as general pattern machines (Mirchandani et al., 2023), they have not been shown to be able to reason over floating point numbers specifically. This is critical in contact-rich settings because they require millimeter level precision. In this section, we thus examine two arithmetic tasks that we believe to be precursors to

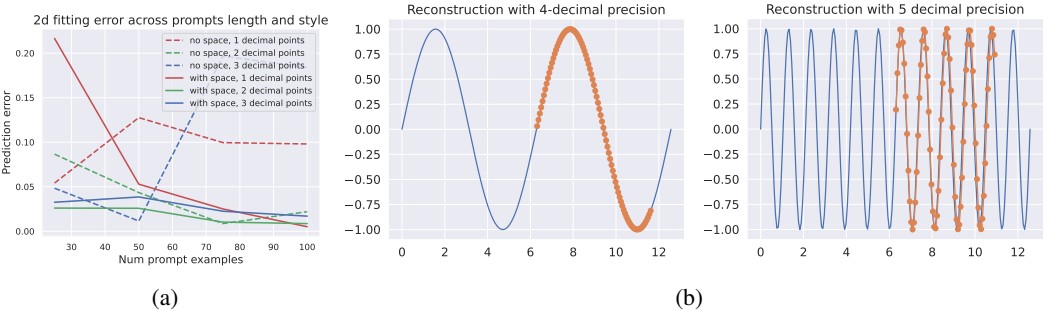

Figure 6: (a) Reconstruction error across different floating point number resolutions and formatting styles. (b) Reconstructions of exemplary sinusoidal sequences for different floating point number resolutions.

reasoning successfully in continuous action spaces: (1) a linear regression task, and (2) a continuous sequence extrapolation task. In both cases, and in contrast to prior work (Mirchandani et al., 2023), we let the LLM operate on floating point numbers directly.

**Zero-Shot Regression In-Context:** In Figure 6a, we show the result of prompting a model with $x, y = f(x)$ from a 2-dimensional linear function $f$. We quantify the ability of the model to implicitly learn the linear function by measuring the $\hat{y}$ output by the model for a given $x$, and plotting the resulting error for different floating point number resolutions. The x-axis denotes the number of x, y pairs provided to the LLM and the y-axis is the average prediction error of $\hat{y}$ for a given value $x$. We compare two different formatting strategies. In the first, the $x, y$ pairs are written out as `f(x)=y`. The second formatting strategy adds spaces between each digit so that each digit is treated as a separate token. For example, `f(1.393)=4.107` is formatted as `f(1.3 9 3)=4.1 0 7`. As can be seen in Fig. 6a, space-formating with GPT-4 is able to successfully regress to the targets within an error of $0.01$ when using at least $3$ decimal places.

**Sequence Extrapolation:** In Figure 6b, we illustrate the ability of an LLM to extrapolate sequences of two-dimensional series of numbers (concretely, sinusoids of different frequencies), as this might transfer to path-following behaviors required in the robotic context. To generate these plots, we feed samples of the sinusoidal function up to a certain value and then auto-regressively sample from the LLM. The sampled points are shown in orange. For visualization purposes, the true function is shown in blue. Qualitatively, we see in the figure that the LLM is indeed capable of extrapolating the chosen sinusoids. Unlike Mirchandani et al. (2023), we find that this ability also works for floating point numbers. To realize these results, we modify the tokenization strategy by placing a space in between each digit. The extrapolation ability begins to break down as the precision and frequency of the sinusoid increase, which can be seen on the right-most side of Figure 6b.

The takeaway from this first set of experiments is that LLMs trained entirely on offline language data can indeed achieve reasonable performance on arithmetic tasks on continuous spaces. This result shows that large language models are capable of performing least squares up to a small degree of error with in-context learning and encourages us to tackle the much more challenging application to fine-grained manipulation in the next section.

## 5.2 CONTACT-RICH MANIPULATION TASKS

Next, we focus on testing the ability of LLMs to generate code for contact-rich manipulation tasks. Section 5.2.1 details the task setup considered in our experiments, while the baselines and method ablations are discussed in Section 5.2.2. We discuss experimental results in Section 5.2.4 and provide further ablation and analysis of the prompting strategies in Section 5.3.

### 5.2.1 TASK DESCRIPTION

**Functional Manipulation Benchmark (FMB):** The Functional Manipulation Benchmark (Luo et al., 2023) studies robotic manipulation, grasping, reorienting, and assembling of a set of dozens of 3D printed objects. The benchmark also emphasizes generalization across different object shapes and positions. We evaluate our approach on a subset of peg insertion tasks across three different object shapes: the circle, star, and half-pipe. We use a script to bring the pegs into a fixed position over the insertion points that includes a randomized rotation around the z-axis. There is no rotation for the circular peg because it has a constant radius. Rotation of the star is sampled uniformly between 0

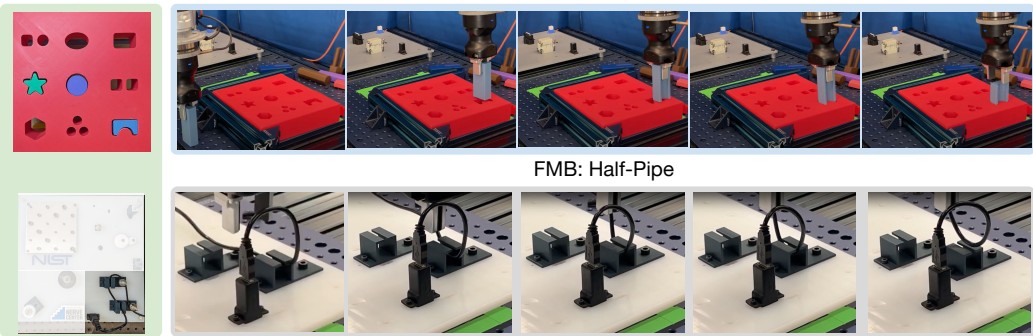

Figure 7: Left: The Functional Manipulation Benchmark (FMB) (Luo et al., 2023) and IROS RGMCS 2020 (Sun et al., 2021) we used for experimentation. Both environments have relatively tight tolerances. Right: Example of the rollouts produced by our method for two tasks. See appendix for rollouts of all tasks.

and $\frac{\pi}{2}$. Rotation of the half pipe is sampled uniformly from either 0 or $\pi$. Inserting these peg shapes successfully requires generating different search patterns based on the object's geometry.

**Industrial Manipulation Tasks:** We study a set of industrial manipulation tasks adapted from IROS RGMCS 2020 (Sun et al., 2021). The benchmark is designed to evaluate proficiency in robotic assembly with an emphasis on small and medium sized parts and deformable objects. We consider the wire routing subtasks from it. Specifically, we study routing (insertion) and unrouting (removal) of a wire through a plastic channel component. Tasks environments are visualized in Fig. 7. Across episodes there is noise in the orientation of the cable within the grasp of the gripper and the tautness of the cable.

We conduct our robot experiments on a Universal Robotics UR5e robot, which is a position-controlled robot with ATI Axia80 force-torque sensor at the wrist. To expose the compliant action space to the language model, we prompt it with the doc-string for a Cartesian admittance move with parameters on stiffnesses, impedances, and constraints in reaching a target pose. The exact prompt is available in the appendix. We add a suffix describing the details of the given task in natural language, optionally including certain keywords about relevant motion patterns when the task setup is ambiguous (e.g., we specify that the peg in the FMB insertion tasks is not aligned, which requires the policy to search for the opening).

### 5.2.2 METHODS CONSIDERED

We compare two classes of methods: a scripted baseline policy authored by an expert and different variants of LLM-generated code using the prompting strategies and control APIs outlined in Sections 4.1 and 4.2, respectively. For LLM-generated code, we distinguish few-shot and zero-shot settings. For the former, the prompt includes examples of the control APIs being called (Fig. 3e). For the latter, the prompt includes Task Description (Fig. 3a), Control API Description (Fig. 3b), and Hint (Fig. 3c). The IROS RGMCS board examples also use the Spatial Pattern prompting strategy (Fig. 3d) for resolving spatial ambiguity in the task. The exact prompts are included in Appendix Section A.

**Scripted [Baseline]:** We compare against a scripted pattern search insertion move that is tuned by an expert on a single task setting. This baseline reflects an alternative to our approach where a single skill is added to our control library, but is not able to be tuned by an expert across different generalization settings. On the Functional Manipulation Benchmark, we adapt a pattern search insertion skill for peg insertion. The scripted move implements fixed get-in-contact, pattern search, and insertion phases, with durations, motion patterns, and force thresholds set by an expert on the circle setting.

**Code-as-Policies (Liang et al., 2022) [Baseline]:** We compare with a baseline approach akin to the prior work (Liang et al., 2022; Mirchandani et al., 2023) that uses the point-to-point action space for performing robot manipulation tasks, i.e. to directly command the robot to move to Cartesian target poses (Fig. 4).

**Ours, Fixed Compliance:** For each task, we compare against a baseline where we do not expose the stiffness and impedance targets or the force constraints to the LLM planner, but instead, use

Table 1: Success rates for the Functional Manipulation Benchmark.

|  | Circle | Star | Half-Pipe |
|---|---|---|---|
| Scripted | 100% | 10% | 0% |
| Code-as-Policies (Liang et al., 2022) (Zero-Shot) | 70% | 0% | 0% |
| Ours, Fixed Compliance (Zero-Shot) | 100% | 70% | 30% |
| **Ours** (Zero-Shot) | 100% | 80% | 50% |

Table 2: Success rates for the IROS RGMCS Industrial Manipulation Benchmark Tasks.

|  | Cable Unroute | Cable Route |
|---|---|---|
| Code-as-Policies (Liang et al., 2022) (Few Shot) | 40% | 0% |
| Ours, Fixed Compliance (Few Shot) | 80% | 30% |
| Ours (Zero-Shot) | 60% | 0% |
| **Ours** (Few-Shot) | 90% | 100% |

predefined compliance parameters. This ablates the importance of force constraints in completing the task, making the action space similar to the one of prior work (Liang et al., 2022), but with compliant motions. Concretely, we provide a modified prompt and access to a wrapper around the Cartesian admittance move that provides fixed stiffness and impedance targets and a fixed constraint on translation error.

**Ours (Few-Shot):** We expose force constraints to the language model and add examples of calls to our control API, which includes conditional compliant moves. This is similar to the Code-as-Policies + Fixed Compliance baselines, but all of the force constraints and termination conditions are exposed to the language model (Fig. 5). In these experiments, each "shot" is an example subcommand that shows how to call the API. In the IROS RGMCS tasks, we include 3 example subcommands: moving down until contact is reached, moving up unless a snag is detected, and moving right unless a snag is detected.

**Ours (Zero-Shot):** We follow the same approach of exposing force constraints to the language model as Ours (Few-Shot, Fig. 5), but do not include any examples of the control APIs being used. This is the most difficult generalization setting because every command is an unseen command.

### 5.2.3 EVALUATION PROTOCOL

Similar to (Yu et al., 2023), we take the best prompt out of 5 samples from the language model and run 10 evaluations. All of our experiments use GPT-4 with a temperature of 0.0 as the underlying LLM. To make comparisons between different action spaces as fair as possible, we take the most successful code generated from our method and overwrite the control API to implement the relevant action space. For each environment, we tune the insertion reference pose that appears in the prompt. Concretely, this is the reference pose used in the make contact in the admittance moves. This hyperparameter is essential on the Point-to-Point baseline because insertion reference poses that are too deep cause a fault.

### 5.2.4 RESULTS

**Functional Manipulation Benchmark.** The main purpose of our first evaluation task is to isolate the ability of different approaches to policy generation to generalize across different task settings, i.e., the ability to generalize insertion search patterns based on different object geometries. The results on the Functional Manipulation Benchmark are listed in Table 1. We find that generating policy code through an impedance action space (Ours) outperforms other methods across different peg shapes. Our baseline scripted policy is successful on the star only when the points are already in close-enough alignment with the hole and fails on the half-pipe shape, which is the most difficult to align because there is only one valid orientation for a successful insertion. In contrast, our method is successful on the half-pipe 50% of the time. Upon further analysis, we notice that this is because the generated code generates a successful policy in only one direction of rotation (i.e., 100% successful for one rotation and 0% successful for the other randomized rotation). When we inspect the code output of the LLM, we find that it generates intuitive waypoints for the search that correspond to the object specified in the prompt. For example, for the half-pipe, the output waypoints oscillate between 0 and $\frac{\pi}{2}$ while for the star shape they go through multiples of $\frac{\pi}{4}$. We also find that the language model does not require examples demonstrating how to use the admittance move, which

is particularly important since the zero-shot prompting setup is more scalable in practical robotics applications.

**Industrial Manipulation Tasks.** Given these results, we proceed to presenting the results of the IROS RGMCS Industrial Manipulation Tasks experiments, which are arguably more directly targeted towards force-based manipulation than the peg insertion task. Indeed, we find it difficult to design an analogous scripted policy baseline for cable (un-)routing that would perform well across both tasks, which is why we omit it in this experiment. From the results in Table 2, we observe that our method (Few-Shot) again consistently outperforms the baselines. Fixed Compliance (Few Shot) is second-best, while Code-as-Policies (Liang et al., 2022) (Few Shot) performs worst, failing to complete the routing task even a single time. Interestingly, the Zero-Shot version of our method also fails for cable routing, for two reasons: (1) the language model tends to generate while loops that are incompatible with the way the API is structured and (2) the program is successful with a much more narrow range of force constraints that are difficult to infer without any more information about the environment. We note that further increasing the performance of LLM-generated policy code on routing would likely require a richer perception API to be made available to the language model.

## 5.3 ABLATING PROMPT HINTS

In the experiments underlying Section 5.2.4, we observed that incorporating additional hints is critical in eliciting relevant motion patterns. To understand this phenomenon in more depth, we plot the distribution over different types of errors for different types of hints for the cable un-routing task in the Zero-Shot setting. We test three hint types: specifically asking for a pattern search, adding extra rules about accessing undefined variables, and asking the model to translate from pseudo-code, similar to Chain-of-Thought prompting (Wei et al., 2022). Figure 8 plots the distribution of failures for combinations of these different types of hints. We perform rejection sampling to estimate the likelihood of each error type given that the generated code fails.

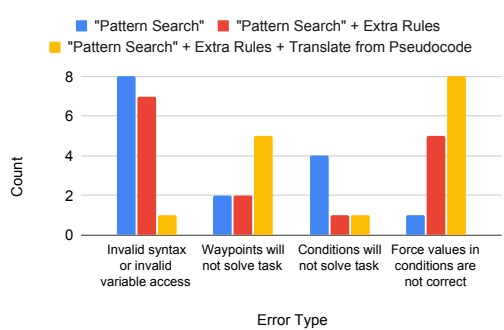

Figure 8: Error Types Across Different Types of Hints.

The errors are ordered from left to right based on the level of intervention required by a human operator to make the task succeed. For example, if the model outputs runnable code, but only moves the cable up without any wiggling motion, the operator will likely need to add new waypoints and tune the termination conditions and force values. The most common example of invalid syntax and variables accesses that we observe is in attempts to access the force values directly to construct for loops over waypoints. Explicitly adding rules against for loops helps somewhat, but asking the model to translate from pseudocode is the most helpful in minimizing these types of errors.

## 6 CONCLUSION

In this work, we study the capabilities of Large Language Models (LLMs) to generate policies for a variety of high-precision contact-rich manipulation tasks in a zero-shot fashion. We find that providing LLMs with the right parameterized action spaces is the key to success, which in this case, correspond to robot impedances and constraints on the interaction forces enabled. We validated our approach on subtasks derived from the Functional Manipulation Benchmark (FMB) and the Robotic Grasping and Manipulation Competition, where zero-shot policy generation in this action space improved success rates over non-compliant action spaces by 3x and 4x, respectively. Our results suggest that LLMs are well-suited for generating code for contact-rich tasks due to their ability to: (1) Recapitulate world knowledge about different motion patterns, and (2) Complete control-relevant arithmetic reasoning tasks over continuous numbers in-context. A slightly surprising result has been that an LLM, intentionally built to output **text**, can actually be used to reason over continuous variables – this domain was normally left to models that were trained to output continuous variables from the very beginning. These results may blur the boundary between use cases that need text output and continuous variable output. In future work, we plan to investigate the use of LLMs for generating policies for more delicate manipulation tasks, including perception APIs and multi-step scenarios.

REPRODUCIBILITY

Readers can reproduce our results by using the prompts from the appendix and included on our paper website. The complete code for the experiments in Figure 6b is also provided on our anonymized paper website: https://dex-code-gen.github.io/dex-code-gen/.

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

## A   PROMPT AND OUTPUT CODE EXAMPLE

In this section, we provide a complete prompt example for the cable un-routing task and the GPT-4-generated output. Importantly, this example demonstrates how that the language model is capable of generating novel combinations of moves and constraint conditions. In the prompt, there is no constraint on right-ward snags for poses moving to the left in the prompt, but the language model is able to synthesize these constraints for the task. Please see our paper website for the remaining prompts.

### A.1   PROMPT EXAMPLE

```python
    prompt = """
You're a robot holding a cable that's threaded through a tunnel with a small opening at the
  top.
You need to unroute the cable by removing it from the tunnel.

You will have access to the following methods, which are imported directly:
- cartesian_admittance_move: This moves the robot to a target_pose until a termination
  condition is reached.
  Args:
      max_cartesian_stiffness:
          The maximum allowed stiffness along each cartesian dof (6d), expressed in
          the robot base frame.
      target_impedance:
          (0,1] 6d-vector specifying the target impedance along each cartesian dof.
      target_pose:
          Target pose for the robot flange frame in the base frame.
      termination_condition:
          Termination condition.
      virtual_cartesian_inertia:
          The diagonal representation of the desired virtual Cartesian inertia
          matrix, expressed in the robot base frame [kg, kg m^2]
      execution_timeout_seconds:
          Timeout for execution. Defaults to 30s if not specified
          Default value: 10.0
      tare_ft_sensor: False when in contact, True otherwise.
- types_pb2.Comparison: this specifies the termination condition above. It can't be accessed
  directly, it can only be passed as an argument to the cartesian_admittance_move method.
  Args:
      operation: types_pb2.Comparison type. One of GREATER_THAN_OR_EQUAL, LESS_THAN_OR_EQUAL,
  APPROX_EQUAL, or APPROX_NOT_EQUAL.
      state_variable_name: Variable where condition is applied. One of
  policy.status.{x_force, y_force, z_force, translation_error, rotation_error}.
      double_value: [-0.4, 0.4] value of state variable on which operation is applied
- types_pb2.Condition: Can't be accessed directly, can only be passed as an argument to the
  cartesian_admittance_move method.
  Args:
      comparison: types_pb2.Comparison type.
- types_pb2.ConjunctionCondition: Wrapper to compose multiple conditions together.
  Args:
      operation: types_pb2.ConjunctionCondition type. One of ALL_OF or ANY_OF .
      conditions: A list of Conditions over which the operation is applied
- pose3_rpy: returns a pose object that specifies a pose. Note: Poses can be composed using
  the multiply method. E.g., pose1.multiply(pose2) applies pose2 on top of pose1.
  Args:
      translation: x, y, z translation.
      rotation: rotation along x, y, and z.
- move_gripper: opens and closes the gripper.
  Args:
      percent_open: Controls how open gripper is. 0 is fully closed, 100 is fully open.
- grasp_cable: grab the free edge of the routed cable

You will also have access to the following fields:
- pose(number): the robot pose just above number. You can build more poses by using this as a
  reference with the multiply method.
  Args:
      number: the pose number for the layout below.
- grasp(pose): go to grasp pose and grasp the cable.
  Args:
      pose: the robot pose above the grasp position

Rules:
- Don't define any new methods
- Don't call any undefined methods
- Don't add any if statements or while loops

Here are some examples using the cartesian_admittance_move method while holding the cable:

```python
position_condition = types_pb2.Comparison(
```

```
  operation=types_pb2.Comparison.LESS_THAN_OR_EQUAL,
  state_variable_name="policy.status.translation_error",
  double_value=0.001,
)

# move down until contact is reached
upward_force_threshold = 0.4
contact_condition = types_pb2.Condition(
    comparison=types_pb2.Comparison(
        operation=types_pb2.Comparison.GREATER_THAN_OR_EQUAL,
        state_variable_name="policy.status.z_force",
        double_value=upward_force_threshold,
    )
)
down_pose = pose(1).multiply(pose3_rpy(translation=[0, 0, -.01]))
cartesian_admittance_move(
  target_pose=down_pose,
  termination_condition=termination_condition=types_pb2.ConjunctionCondition(
        operation=types_pb2.ConjunctionCondition.ANY_OF,
        conditions=[contact_condition, position_condition]
    ))

# move up unless snag is detected
downward_force_threshold = -0.4
snag_condition = types_pb2.Condition(
    comparison=types_pb2.Comparison(
        operation=types_pb2.Comparison.LESS_THAN_OR_EQUAL,
        state_variable_name="policy.status.z_force",
        double_value=downward_force_threshold,
    )
)
up_pose = pose(1).multiply(pose3_rpy(translation=[0, 0, .01]))
cartesian_admittance_move(
  target_pose=up_pose,
  termination_condition=termination_condition=types_pb2.ConjunctionCondition(
        operation=types_pb2.ConjunctionCondition.ANY_OF,
        conditions=[snag_condition, position_condition]
    ))

# move the cable to the right unless snag is detected
leftward_force_threshold = 0.4
leftwardsnag_condition = types_pb2.Condition(
    comparison=types_pb2.Comparison(
        operation=types_pb2.Comparison.GREATER_THAN_OR_EQUAL,
        state_variable_name="policy.status.x_force",
        double_value=leftward_force_threshold,
    )
)
right_pose = pose(1).multiply(pose3_rpy(translation=[-0.01, 0, 0]))
cartesian_admittance_move(
  target_pose=up_pose,
  termination_condition=termination_condition=types_pb2.ConjunctionCondition(
        operation=types_pb2.ConjunctionCondition.ANY_OF,
        conditions=[leftwardsnag_condition, position_condition],
    ))
```

Currently the robot is holding the cable at pose(1), but the cable is stuck in a tunnel with
  a small opening at the top. Describe a strategy for unrouting the cable using force
  information to search for the opening and then write a program using the APIs above to
  unroute the cable and lift it out of the tunnel."""
```
messages = [{"role": "user", "content":prompt}]
response = openai.ChatCompletion.create(
    model="gpt-4",
    messages = messages,
    temperature=0.0,
    max_tokens=512*3,
    frequency_penalty=0.0
)
print(response["choices"][0]["message"]["content"])
```

## A.2 GENERATED CODE EXAMPLE

```
    Strategy:
1. First, we will move the cable upwards until we detect a snag. This will help us to find
  the top of the tunnel.
2. Once we detect a snag, we will move the cable to the right until we detect another snag.
  This will help us to find the right wall of the tunnel.
3. After finding the right wall, we will move the cable to the left until we no longer detect
  a snag. This will help us to find the opening of the tunnel.
4. Finally, we will move the cable upwards again to lift it out of the tunnel.
```

```python
Program:

```python
# Step 1: Move up until snag is detected
downward_force_threshold = -0.4
snag_condition = types_pb2.Condition(
    comparison=types_pb2.Comparison(
        operation=types_pb2.Comparison.LESS_THAN_OR_EQUAL,
        state_variable_name="policy.status.z_force",
        double_value=downward_force_threshold,
    )
)
up_pose = pose(1).multiply(pose3_rpy(translation=[0, 0, .01]))
cartesian_admittance_move(
  target_pose=up_pose,
  termination_condition=types_pb2.ConjunctionCondition(
        operation=types_pb2.ConjunctionCondition.ANY_OF,
        conditions=[snag_condition, position_condition]
    ))

# Step 2: Move right until snag is detected
leftward_force_threshold = 0.4
leftwardsnag_condition = types_pb2.Condition(
    comparison=types_pb2.Comparison(
        operation=types_pb2.Comparison.GREATER_THAN_OR_EQUAL,
        state_variable_name="policy.status.x_force",
        double_value=leftward_force_threshold,
    )
)
right_pose = pose(1).multiply(pose3_rpy(translation=[-0.01, 0, 0]))
cartesian_admittance_move(
  target_pose=right_pose,
  termination_condition=types_pb2.ConjunctionCondition(
        operation=types_pb2.ConjunctionCondition.ANY_OF,
        conditions=[leftwardsnag_condition, position_condition],
    ))

# Step 3: Move left until no snag is detected
rightward_force_threshold = -0.4
rightwardsnag_condition = types_pb2.Condition(
    comparison=types_pb2.Comparison(
        operation=types_pb2.Comparison.LESS_THAN_OR_EQUAL,
        state_variable_name="policy.status.x_force",
        double_value=rightward_force_threshold,
    )
)
left_pose = pose(1).multiply(pose3_rpy(translation=[0.01, 0, 0]))
cartesian_admittance_move(
  target_pose=left_pose,
  termination_condition=types_pb2.ConjunctionCondition(
        operation=types_pb2.ConjunctionCondition.ANY_OF,
        conditions=[rightwardsnag_condition, position_condition],
    ))

# Step 4: Move up to lift the cable out of the tunnel
cartesian_admittance_move(
  target_pose=up_pose,
  termination_condition=types_pb2.ConjunctionCondition(
        operation=types_pb2.ConjunctionCondition.ANY_OF,
        conditions=[snag_condition, position_condition]
    ))
```
```

## B  PROMPT ABLATION DETAILS

To generate Figure 8, we take the first 15 prompts that generate an error (skipping any successful prompts) and classify them based on failure type. Table 2 uses the best of five samples from the complete prompt.

## C  COMPLIANT ACTION SPACE PARAMETERIZATIONS

It is worth noting that different robotic platforms offer different interfaces to controlling their robots. Torque controlled robots, such as Franka-Emika Panda and Kuka LBR iiwa robots, readily lend themselves to the impedance control law described in Equation 1. On the other hand, position controlled robots (like the UR5e used in our experiments) require a target position.

To achieve soft interaction with the environment, we can implement an admittance controller that makes use of a force-torque sensor attached to our robot end-effector:

$$\ddot{x} = M^{-1}(K_p(x_d - x) + K_d(\dot{x}_d - \dot{x}) + (f_d - f_s))$$

where $f_d$ is a desired force value, and $f_s$ is the current sensed force value. The use of an admittance controller enabled us to parameterize our action space as $t_i = ([\mathbf{x}_{target}]_i, \sigma_i)$, where $[\mathbf{x}_{target}]_i$ denote target Cartesian pose and $\sigma_i$ denote the stiffness vector.

Note that even though the admittance controller differs from the impedance controller described in Equation 1, the parameters, namely $K_p$, $K_d$, and $M$ play a conceptually similar role to their corresponding variables in the impedance control law, and so we believe our method can generalize across these different robotic platforms.

## D  ADDITIONAL RESULTS

We visualize rollouts from our method in Figure 9.

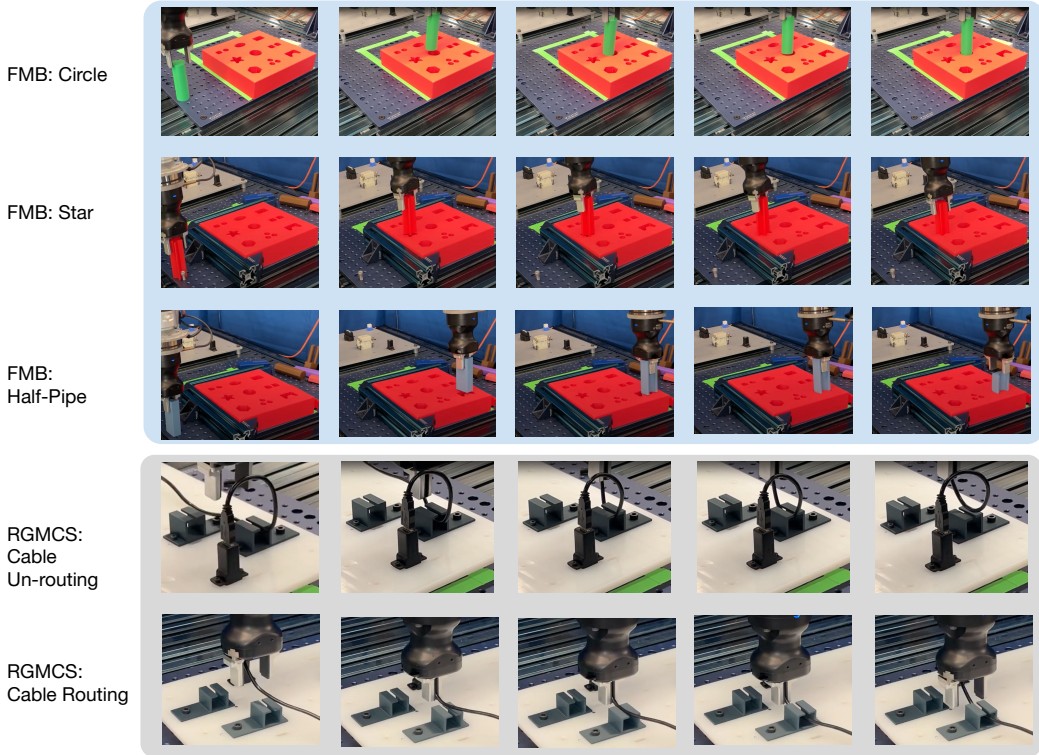

Figure 9: Example rollouts on the experimental tasks.

