# OpenReview forum: "Generating Robot Policy Code for High-Precision and Contact-Rich Manipulation Tasks"
_ICLR.cc/2024/Conference — Submitted to ICLR 2024_

### Official Review · Reviewer_dGpr · 2023-10-31

**Soundness:** 3 good
**Presentation:** 4 excellent
**Contribution:** 2 fair
**Rating:** 5
**Confidence:** 5

**Summary:**

This paper is about using LLMs for contact rich manipulation tasks. Prior methods use LLMs to come up with a series of function calls in the position space. This paper rightfully uses impedance control for contact rich tasks. The actions are represented by a combination of target position and also the stiffness vector along each axis. In my opinion using admittance control is extremely important for this contact rich task. Without admittance control, none of the prompting matters. The prompt has a task description in plain English, list of functions along with their params, hints, spatial patterns (which I have not seen to be used in the example prompts), and some examples to show how to use particular functions specially the admittance control functions. In general, I think this paper is a step in the right direction but has its limitations.

**Strengths:**

- Contact rich manipulation is a very important area in robotics.
- Using LLM for contact rich task is an important area and I think this paper is the first to do it.
- The paper is well-written and easy to follow.
- I like the ablation studies that show the importance of impedance control, position control, and impedance control (fixed) which shows the importance of predicting the values for impedance control.

**Weaknesses:**

- On one hand, the paper is very similar to code-as-policy and other variants. On the other hand, it is known that impedance control is crucial for contact-rich tasks.
- Even though the paper emphasizes on continuous aspect of the prediction, there is not much continuous quantities going on in the predicted codes. The values of floats in the generated code does not have much variation. The paper gives the impression that it can predict full range of continuous values but in the examples the continuous values can be looked at as a value drawn from a very small set such as `{0, 0.01, 0.02, 0.04, 0.001}`.
- The part that really needs continuous value is abstracted away as `pose(1)` and the model just apply relative transforms to that. If `pose(1)` is chosen close to the final poses, the LLM does not need to do much to accomplish the task. It would help if the paper reports the distribution of `pose(1)` distance to the final target poses.
- Not all the prompt categories are used in the example prompt. I am particularly interested to see a prompt that uses Spatial Patterns. I also believe that this is a simplistic prompt. The resolution of patterns can make a big difference in success or failure of tasks. In addition, how are the continuous orientation of parts represented by characters?
- Section 5.1 is interesting but not sure how it connects to the rest of the paper. I don’t see the use of spacing between digits in the example prompts. In addition, none of the prompts have numbers beyond just double floating point precision. They don't even utilize all the possibilities in the double digit floating points (referring to my earlier point about floating numbers are coming from a set of few floating number which can be represented
- The paper assumes that perception is a solved problem and it does not deal with uncertainties in the perception.
- What is total number of trials for Table 1 and 2?

**Questions:**

See weaknesses section.

After authors' responses and thinking more, I decided to keep my current score.

---

> ### Author Response · Authors · 2023-11-16
> **Response to Reviewer dGpr (1/1)**
>
> 1. _Even though the paper emphasizes on continuous aspect of the prediction, there is not much continuous quantities going on in the predicted codes. … The part that really needs continuous value is abstracted away as pose(1) and the model just apply relative transforms to that. … Section 5.1 is interesting but not sure how it connects to the rest of the paper. I don’t see the use of spacing between digits in the example prompts._
>
> We realized that this was indeed unclear in the paper and we'll fix it in the following way: the goal of our initial experiment was to probe the capabilities of LLMs for high-precision prediction and address a shortcoming present in prior work [1]. We are working on demonstrating the importance of these probing experiments in performing high-precision tasks in real-world robotics scenarios during the rebuttal period.
>
> 2. _The part that really needs continuous value is abstracted away as pose(1) and the model just apply relative transforms to that. If pose(1) is chosen close to the final poses, the LLM does not need to do much to accomplish the task. It would help if the paper reports the distribution of pose(1) distance to the final target poses._
>
> The choice of pose is kept constant across all baselines. In spite of this, the task is difficult to achieve without the correct action space because successful routing and unrouting requires reasoning over interaction forces. This is why the point-to-point baseline is only 20% successful on average across cable routing tasks. Even with a fixed pose, there is randomization in the orientation of the cable at the point of grasping.
>
> 3. _Not all the prompt categories are used in the example prompt. I am particularly interested to see a prompt that uses Spatial Patterns. I also believe that this is a simplistic prompt. The resolution of patterns can make a big difference in success or failure of tasks. In addition, how are the continuous orientation of parts represented by characters?_
>
> That is correct. We are working on adding examples using the spatial patterns prompt during the rebuttal period. The main purpose of this prompt is to relay high level spatial information about the relative position of objects in the scene. It doesn’t contain information about the continuous orientation of parts represented by the characters.
>
> 4. _The paper assumes that perception is a solved problem and it does not deal with uncertainties in the perception._
>
> We agree that incorporating perception is an important step for future work and have added this into the conclusion.
>
> 5. _What is total number of trials for Table 1 and 2?_
>
> Following the protocol of VoxPoser [2] in Table 1, we report the average success over 10 evaluation runs. We report results on the best of 5 generated code examples. We’ve highlighted this experimental detail in the revised PDF: see the first line of section 5.2.3.
>
> [1] Mirchandani et. al. Large Language Models as General Pattern Machines. CoRL 2023.
>
> [2] Huang et. al. VoxPoser: Composable 3D Value Maps for Robotic Manipulation with Language Models. https://arxiv.org/pdf/2307.05973.pdf. CoRL 2023 (Oral).

---

> ### Author Response · Authors · 2023-11-22
> **New spatial prompt example on website**
>
> Hi reviewer dGpr,
>
> Thank you again for your constructive comments and feedback.
>
> We've added an example with a spatial prompt for a cable unrouting task to our website (at the bottom): https://dex-code-gen.github.io/dex-code-gen/. We take the _same_ prompt used for the cable routing and unrouting tasks and append the new task description as well as a spatial summary of the board layout:
>
> ```
> Now assume the board looks like this from the top down:
>     B
> ccc c
> cscsc
> c ccc
> 1
>
> x translations move left to right and y translations moves bottom to top. The distance between screws is .03M.
>
> where s is a screw, c is the path of the cable and B is the base that the cable is attached to. Explain how to unravel the cable from the screws and then generate the code to unravel the cable if the robot is currently holding the cable at position 1.
> ```
>
> Please let us know if you feel some of your concerns are still unaddressed.

---

> > ### Comment · Reviewer_dGpr · 2023-12-04
> > **Post Rebuttal**
> >
> > I thank the authors on responding to the questions. I am not convinced that for contact rich tasks separation of vision and controller is really practical. For contact rich tasks there needs to be a reactivity to forces that is beyond just impedance control and pose estimation.
> >
> > I strongly recommend doing it yourself with hands and eyes cause this paper really inspired me to see if it's a valid approach or not. Try to use screwdriver with screw. Even with human level vision, you won't put screwdriver exactly on the middle of the screw consistently. Quite often the screwdriver ends up a bit off from the screw center. The adjustment comes after making initial contact. I think if the paper would've taken the route of both vision and control, these challenges would've been more obvious. I encourage authors consider scenarios where there needs to be adjustments after making the initial contact.
> >
> > On the other hand, I somewhat agree with the authors that this is a first work in this area but I think the method still has to have some decent performance. Best of 5 results opens the door for cherry picking and not necessarily having a working method. Maybe pick fewer tasks but show repeatable success instead of going for a lot of tasks and then use best of 5 as success.

---

### Official Review · Reviewer_nuUX · 2023-11-01

**Soundness:** 3 good
**Presentation:** 3 good
**Contribution:** 2 fair
**Rating:** 5
**Confidence:** 4

**Summary:**

Authors tackled contact-rich robotic control tasks by taking advantage of LLMs. As opposed to most previous efforts that operated on higher level abstraction, this paper focuses on lower-level control. Additionally, authors allowed the LLM to also generate constraints to enable closed-loop control. Empirical results on peg-in-the-whole and (un)route cable domains showed up to 20% and 70% success rate improvement compared to best baseline techniques respectively.

**Strengths:**

+ Simplicity: The approach is easy to adopt, as the main planning component is the readily available GPT4. The trick to enter decimal points through a sequence of space separated tokens was intriguing.
+ Empirical Results: the improvement in task success rate looks promising
+ Writing: the paper was easy to follow

**Weaknesses:**

- Claims: the 3x and 4x improvement was a bit oversold. I recommend authors to focus on the improvement over the best alternative approach rather than the point to point.
- Novelty: The paper introduces limited novelty compared to the previous work. The main contribution is to allow LLMs to also generate constraints for their manipulation functions.
- Generalizability: given the few-shot learning example in the appendix, the prompt holds key sections for solving the task (e.g. move up unless snag is detected). It would have been great if authors included only examples not exactly needed to solve the task for the second domain. We see with no examples, LLM did not do great specially on the harder task of routing the cable.
- Lack of details: some key details of the experimentations were missing (see questions)

Minor comments:
the Transformers => Transformers

**Questions:**

- Figure 6 a,b: For each point generated by LLM, is the input true history, or is it based on previous points generated by LLM. My understanding if the former, but would be great to clear this out in the paper.
- "every command is an unseen command.", I think you meant unseen in the sense of not seeing examples of it but the command is defined as an input through the prompt and hence seen.
- "We also tune the following properties independently for each model". How did you optimize them?
- Figure 8: While the type of errors made by LLM changed as more hints were introduced the total number of errors seemed to remain constant. This means the hint type did not change the success rate of the agent. Is that true? If so, please make it explicit it in the paper.
- Can you provide details on the number of shots used for few-shot experiments?

---

> ### Author Response · Authors · 2023-11-16
> **Response to Reviewer nuUX (1/2)**
>
> 1. _Claims: the 3x and 4x improvement was a bit oversold. I recommend authors to focus on the improvement over the best alternative approach rather than the point to point._
>
> We’ve modified that abstract to make this distinction more clear. To clarify, the point-to-point method is most analogous to prior work such as Code-as-Policies. The best alternative approach (Fixed Compliance) is an ablation of our proposed action space that does not include termination constraints or modifiable compliance parameters. Generating robot policy code with Fixed Compliance has not been tried in prior work; we are the first to do so. In the updated PDF, we have changed the names of the baselines and ablations to make this more clear.
>
> 2. _Novelty: The paper introduces limited novelty compared to the previous work. The main contribution is to allow LLMs to also generate constraints for their manipulation functions._
>
> It has been long discussed in the robotics community whether LLM advancement could ever result in contact-rich dexterous control and this is the first paper showing signs that this is indeed possible. We develop a system for automatically generating robot policy code for dexterous tasks by allowing LLMs to specify constraints on the stiffness, forces, and trajectories required to perform contact-rich manipulation tasks and show that our approach outperforms the state-of-the-art approaches such as Code-as-Policies by a significant margin, which uses a contact-unaware model.
>
> 3. _Generalizability: given the few-shot learning example in the appendix, the prompt holds key sections for solving the task (e.g. move up unless snag is detected). It would have been great if authors included only examples not exactly needed to solve the task for the second domain. We see with no examples, LLM did not do great specially on the harder task of routing the cable._
>
> We agree that adding a targeted generalization study could help improve the overall quality of the paper and are working to add in more experiments that demonstrate this by the end of the rebuttal period. Our finding that the zero-shot prompt performs poorly is consistent with prior work on LLMs and robotics. Most work in this area uses a large number of prompt examples. For example VoxPoser [1] leverages 8 different prompts with an average of ~9 subcommands per prompt. See an example here: https://github.com/huangwl18/VoxPoser/blob/main/src/prompts/rlbench/composer_prompt.txt. For code generation benchmarks, Code-As-Policies [2] uses a prompt with 24 pre-defined functions: https://github.com/google-research/google-research/blob/master/code_as_policies/Experiment_%20Robot%20Code-Gen%20Benchmark.ipynb. Our main goal in this experiment is to test the composition of primitives in solving a cable routing task, which is why we provide 3 sub-commands in the same order for each task. The LLM successfully generates compositions of the subcommands to solve the task. For the cable-unrouting task, the generated code also includes newly defined constraints that detect rightward force. This example is present on the paper website and we’ve updated the appendix to include this as well.
>
> 3. _Figure 6 a,b: For each point generated by LLM, is the input true history, or is it based on previous points generated by LLM. My understanding if the former, but would be great to clear this out in the paper._
>
> In Figure 6b, all of the visualized orange points are generated from previous points generated by the LLM. In Figure 6a, the x-axis represents the number of true pairs input and the y-axis represents the average prediction error of the next predicted y value. We have updated the draft to make this clear.
>
> 4. _"every command is an unseen command.", I think you meant unseen in the sense of not seeing examples of it but the command is defined as an input through the prompt and hence seen._
>
> Yes. To be specific, the command is the last section of the prompt. We do not provide paired code examples for the given command.
>
> 5. _"We also tune the following properties independently for each model". How did you optimize them?_
>
> We’ve updated Section 5.2.3 to be more precise. The insertion depths are manually tuned by a human operator once for a given environment and then specified in the prompt. This is to make the Point-to-Point baseline as strong as possible when comparing against our method. Otherwise, the Point-to-Point policy will cause the robot to fault at greater insertion depths because it will continue to apply force once in contact.
>
> [1] Huang et. al. VoxPoser: Composable 3D Value Maps for Robotic Manipulation with Language Models. https://arxiv.org/pdf/2307.05973.pdf. CoRL 2023 (Oral).
>
> [2] Liang et. al. Code as Policies: Language Model Programs for Embodied Control. https://code-as-policies.github.io/. CoRL 2022.

---

> ### Author Response · Authors · 2023-11-16
> **Response to Reviewer nuUX (2/2)**
>
> 6. _Figure 8: While the type of errors made by LLM changed as more hints were introduced the total number of errors seemed to remain constant. This means the hint type did not change the success rate of the agent. Is that true? If so, please make it explicit it in the paper._
>
> In Figure 8, we perform rejection sampling to estimate the likelihood of each error type given that the generated code fails. The hint type does change the success rate of the agent. We have updated the paper to be more clear.
>
> 7. _Can you provide details on the number of shots used for few-shot experiments?_
>
> The prompt contains 3 examples where the control API is called with relevant subcommands: moving down until contact is reached, moving up unless a snag is detected, and moving right unless a snag is detected. We count each example subcommand as a single shot. Please see the updated Section 5.2.2 where this is clarified.

---

> ### Author Response · Authors · 2023-11-22
> **New results on generalizing the same prompt**
>
> Hello, to demonstrate the generalizability of our approach, we've added another experiment where we take the _same_ prompt (e.g., move up unless snag is detected) examples as the cable route and unroute tasks and change the user command to perform a connector insertion task. This experiment shows that the examples provided in the prompt are general and that a non-trivial number of tasks can be solved by composing together these examples. Indeed, when we run this task we see that our approach is still strong:
>
> |   | Connector Insertion |
> |---|---|
> | **Code-as-Policies (Liang et al., 2022) (Few Shot)** | 0% |
> | **Ours, Fixed Compliance (Few Shot)** | 20% |
> | **Ours (Zero-Shot)** | 0% |
> | **Ours (Few-Shot)** | 90% |
>
> The exact prompt, output, and a video of a policy rollout are available on our website: https://dex-code-gen.github.io/dex-code-gen/.
>
> Thank you for dedicating your time and effort to evaluate our work. Please let us know if you have any unresolved concerns with our work.

---

> > ### Comment · Reviewer_nuUX · 2023-12-05
> >
> > I would like to thank authors for additional information and also trying a new example to better gauge the generalizability of their approach. The new connector insertion task relies mostly on a grid search for which the LLM has a good understanding, as shown in other examined tasks. One good path to test generalizability is to do the cable routing without providing key sections  of the code as highlighting in my question. Overall I think the work is very interesting but still needs maturity before publication. Hence I hold my score.

---

### Official Review · Reviewer_7Rys · 2023-11-02

**Soundness:** 3 good
**Presentation:** 3 good
**Contribution:** 3 good
**Rating:** 5
**Confidence:** 4

**Summary:**

This paper investigates whether the GPT-4 large language model (LLM) can generate scripted policies for contact-rich manipulation tasks. A robot motion API with compliant motions and stop conditions is available to the LLM. The LLM is also given a description of the API, a description of the task state, and optionally a few examples of API usage.

The paper first demonstrates the use of LLMs for floating point input/output with tokenization enforced by putting a space between each digit. Then it demonstrates the success rates of policies suggested by the LLM for insertion and cable unrouting on a real robot.

In general, the paper focuses on convincing the reader that LLMs _are_ capable of generating reasonable instructions for contact-rich tasks in mathematical / code format. It does _not_ focus on convincing that such policies are better than other alternatives like policies learnt by imitation learning, reinforcement learning, heuristic design, etc.

**Strengths:**

- Thoughtful design of the action space (conditional compliant moves) to enable successful completion of contact-rich manipulation tasks by policies generated by LLMs.
- Validation on a real robot setup.
- Clearly articulating the research question "demonstrate that LLMs, without any specialized training, have the ability to perform contact-rich tasks when given the appropriate action space", and conducting experiments to answer it.

**Weaknesses:**

- The paper does not demonstrate LLM policies are better than other policies, because it does not compare against those baselines. In this context, the paper lacks discussion about _why_ LLM policies might be preferable.
  - Section 1 mentions "scaling". Does this refer to scaling in terms of the variety of tasks? Maybe the LLM provides zero-shot generalization abilities to many different tasks. In that case, the experimental validation presented here is quite weak, it only presents two kinds of tasks. Experiments for other contact-rich tasks from previous works, like inserting plates into dishwasher racks [1], loading bookshelves, opening door handles [4], screwing nuts on bolts [5], USB insertion [2], plug insertion [2] etc. would have helped demonstrate the unique scaling ability provided by LLMs.
  - If some other reasons make LLM policies more attractive, please discuss those reasons.
- The impact of hint inputs (Section 4.1(3)) is not experimentally validated. This is important, because from the text it seems like the hints provide strong policy guidance like "do a grid pattern search".
- Impact of noise in state description provided to the LLM (e.g. imperfectly sensed spatial pattern of the board in Fig. 3(d), or noisy robot starting pose) is not examined experimentally.
- Is the scripted policy for the insertion task unfairly weak compared to the LLM? It is tuned to the easiest object `circle` and expected to generalize to `star` and `half-pipe`. On the other hand, the LLM policy is provided with a hint about which object it is planning the policy for.


### References
1. "Zero-Shot Transfer of Haptics-Based Object Insertion Policies" - ICRA 2023
2. "Deep reinforcement learning for industrial insertion tasks with visual inputs and natural rewards" - IROS 2020
3. "kpam 2.0: Feedback control for category- level robotic manipulation" - RA-L 2021
4. "Learning force control policies for compliant manipulation" - IROS 2011
5. "Factory: Fast contact for robotic assembly" - RSS 2022

**Questions:**

- Please provide the policies generated by the expert scripter and the LLM for insertion task. This will allow readers to compare the LLM output to the human-designed scripted policy. And also allow the reader to know which hints were required by the LLM.
- Please clarify the evaluation protocol, because the language is unclear. Especially the last sentence of Section 5.2.3.
- Please explain how the potential generalization advantages of LLM policies are shown in the current experiments.

---

> ### Author Response · Authors · 2023-11-16
> **Response to Reviewer 7Rys (1/2)**
>
> 1. _The paper does not demonstrate LLM policies are better than other policies, because it does not compare against those baselines. In this context, the paper lacks discussion about why LLM policies might be preferable._
>
> Compared to policy learning approaches that rely on reinforcement learning or imitation learning, our approach is far more data efficient and requires less time from human operators. We explain these differences concretely below using some of the references you provided (thank you for bringing them to our attention). Please also see the updated related work where we add these comparisons.
>
> - **Imitation learning:** in the contact-rich push-T task, Diffusion Policies [1] show generalization across the initial position of the T block across 136 demonstrations. Before the data collection can begin  “the operator has [already] performed this task for many hours” to reach the proficiency needed for demonstrations of a sufficient quality.
> - **Reinforcement learning:** in "Zero-Shot Transfer of Haptics-Based Object Insertion Policies" the pre-training phase requires building a dedicated simulator and ~9M simulated environment steps (from Table III of their paper: 64K random steps to seed training + [64K training iterations x 140 steps per training iteration]). "Deep reinforcement learning for industrial insertion tasks with visual inputs and natural rewards" requires 2.5k-10k real world timesteps per task. "Factory: Fast contact for robotic assembly" also requires a dedicated simulator and >2k environment steps to reach success in simulation and is not validated in the real world.
>
> By contrast, our experiments only require tuning the insertion depth specified within the prompt and a natural language command to request a new task. To draw a fair comparison to reinforcement learning approaches, we’d need to restrict the setting to ~3 demonstrations, which would result in 0% success rate. Our results on the Functional Manipulation Benchmark show that one generalization benefit to using a LLM is that we can leverage the language model’s world knowledge about different object types to generalize in a zero-shot fashion. We are working on adding more NIST-style insertion experiments that you described to further demonstrate task generalization.
>
> [1] Chi et. al. Diffusion Policy: Visuomotor Policy Learning via Action Diffusion. RSS 2023.
>
> 2. _The impact of hint inputs (Section 4.1(3)) is not experimentally validated._
>
> We will work on adding an ablation of hint inputs. We will try to add these by the end of the rebuttal period.
>
> 3. _Impact of noise in state description provided to the LLM (e.g. imperfectly sensed spatial pattern of the board in Fig. 3(d), or noisy robot starting pose) is not examined experimental_
>
> We agree that a precise analysis of the impact of noise is important and can work to study this in the future. Currently, there are two sources of noise in our experiments. In the insertion experiments, the star-shaped peg’s initial position is sampled uniformly from 0 to pi/2 and the half-pipe shaped peg is sampled uniformly from 0 or pi. In the cable manipulation, there is noise in the orientation of the cable within the grasp of the gripper and the tautness of the cable varies across episodes. This occurs from minor displacements of the untethered cable at the start of an evaluation episode.
>
> 4. _Is the scripted policy for the insertion task unfairly weak compared to the LLM? It is tuned to the easiest object circle and expected to generalize to star and half-pipe. On the other hand, the LLM policy is provided with a hint about which object it is planning the policy for._
>
> It is correct that we tune the insertion parameters on the circle example. Our goal with this experiment is to show that expert policies require task-specific tuning and don’t work out-of-the-box when the task parameters change. By contrast, LLM-generated policy code only requires a natural language task description.
>
> 5. _Please provide the policies generated by the LLM for the insertion task._
>
> Please see the updated website (https://dex-code-gen.github.io/dex-code-gen/).

---

> ### Author Response · Authors · 2023-11-16
> **Response to Reviewer 7Rys (2/2)**
>
> 6. _Please clarify the evaluation protocol, because the language is unclear. Especially the last sentence of Section 5.2.3._
>
> Thank you for pointing this out. Because the Point-to-Point model has no compliance, it will continue pushing down to make contact until a fault is reached. For example, our prompt contains example pose translations to make contact with the surface, such as `Pose3(translation=(0, 0, -0.02))` where 0.02 is the insertion depth. The compliant policy will be robust to different insertion depths, but the Point-to-Point policy will cause the robot to fault at greater insertion depths because it will continue to apply downward force once in contact. We tune the insertion depth to make the Point-to-Point baseline as strong as possible when comparing against our method. We have updated Section 5.2.3 to be more precise. Please see the updated PDF for this change.
>
> 7. _Please explain how the potential generalization advantages of LLM policies are shown in the current experiments._
>
> In the FMB experiments in table 1, we see that our approach is able to generalize across different object geometries with only high level object descriptions provided to the LLM. By contrast, an expert scripted policy does not work out of the box across geometries. In the RGMCS results, we find that the LLM is able to generate a performant policy for two different cable manipulation tasks. The only change to the prompt across each scenario is the final user command specified.

---

### Official Review · Reviewer_ZcBU · 2023-11-10

**Soundness:** 1 poor
**Presentation:** 1 poor
**Contribution:** 1 poor
**Rating:** 1
**Confidence:** 4

**Summary:**

This paper presents a zero-shot way to perform contact-rich planning for robotic manipulation tasks using LLMs. Overall, I find the paper to be weak and not fit for publication at ICLR.

**Strengths:**

The paper attempts to look at an important problem which can simplify training of robots for a lot of contact-rich tasks.

**Weaknesses:**

The authors need to improve several things:
1. Overall, I feel that the authors fail to capture a principled approach to present the key elements for the contact-rich task. The authors propose to replace the search-based methods for contact-rich tasks by replacing the search with LLM by providing some rules to LLM for decision making. While this might replace the search-based methods, but the proposed method is not general enough to work.
2. Most of contact-rich manipulation tasks end up to be partially observable due to contact formations and unobservability of the contact states. This aspect of the problem can not be handled in the proposed method. This makes the proposed method a rather ad-hoc attempt at solving the problem than a principled way of incorporating LLM for decision making during contact-rich tasks.
3. From a presentation point of view, the authors need to clarify what their decision variables are which are exposed to the LLM. Having that specified will help with understanding about the reasoning performed by LLMs.
4. In the current presentation, the paper appears as a poor, ad-hoc attempt at incorporating LLM for contact-rich tasks.

**Questions:**

The following points are not clear to me:
1.  From a presentation point of view, the authors need to clarify what their decision variables are which are exposed to the LLM. Having that specified will help with understanding about the reasoning performed by LLMs.
2. Is the impedance the only parameter exposed to the LLM?
3. How is the LLM tuning the impedance parameter? Are you allowing LLM to interact with the current system? or its just done using a rule that LLM comes up with?

---

> ### Author Response · Authors · 2023-11-16
> **Response to Reviewer ZcBU (1/1)**
>
> 1. _Most of contact-rich manipulation tasks end up to be partially observable due to contact formations and unobservability of the contact states. This aspect of the problem can not be handled in the proposed method. This makes the proposed method a rather ad-hoc attempt at solving the problem than a principled way of incorporating LLM for decision making during contact-rich tasks._
>
> We agree that incorporating more perception is an important step for future work. We’ve highlighted this discussion in the conclusion of the updated PDF. We find that for these tasks our approach is capable of generating performant policies from force readings alone.
>
> 2. _From a presentation point of view, the authors need to clarify what their decision variables are which are exposed to the LLM. Having that specified will help with understanding about the reasoning performed by LLMs. Is the impedance the only parameter exposed to the LLM?_
>
> We’ve updated our website  (https://dex-code-gen.github.io/dex-code-gen/) with exact prompts and generated code examples to clarify which action spaces are made available to the LLM.
>
> 3. _How is the LLM tuning the impedance parameter? Are you allowing LLM to interact with the current system? or its just done using a rule that LLM comes up with?_
>
> Please find an example of the admittance move doc-string below. This doc-string is passed to the LLM as part of the prompt.
> ```
> """
> - cartesian_admittance_move: This moves the robot to a target_pose until a termination condition is reached.
>   Args:
>       max_cartesian_stiffness:
>           The maximum allowed stiffness along each cartesian dof (6d), expressed in
>           the robot base frame.
>       target_impedance:
>           (0,1] 6d-vector specifying the target impedance along each cartesian dof.
>       target_pose:
>           Target pose for the robot flange frame in the base frame.
>       termination_condition:
>           Termination condition.
>       virtual_cartesian_inertia:
>           The diagonal representation of the desired virtual Cartesian inertia
>           matrix, expressed in the robot base frame [kg, kg m^2]
>       execution_timeout_seconds:
>           Timeout for execution. Defaults to 30s if not specified
>           Default value: 10.0
>       tare_ft_sensor: False when in contact, True otherwise.
> """
> ```
> The LLM outputs policy code that parametrizes this method. For example:
> ```
> cartesian_admittance_move(
>         max_cartesian_stiffness=[1000, 1000, 1000, 300, 300, 300],
>         target_impedance=[0.1, 0.1, 0.1, 0.1, 0.1, 0.1],
>         target_pose=target_pose,
>         termination_condition=search_termination_condition,
>         virtual_cartesian_inertia=[1, 1, 1, 1, 1, 1],
>         execution_timeout_seconds=10.0,
>         tare_ft_sensor=False
>     )
> ```
> We’ve updated the paper website to include more examples of this process.

---

> ### Comment · Reviewer_ZcBU · 2023-11-17
> **rebuttal response**
>
> I would like to thank authors for posting a response. This is what I think is a weak point of the paper is. You treat the contact rich problem as a search problem by using an appropriate impedance parameters. However, consider the problem of Peg-in-hole. This problem tends to be a partially observable problem (and so are numerous other contact-rich problems). What this means a single force reading can not reveal the perfect mis-alignment between the peg and hole. The spiral search methods were proposed decades earlier and since then there has been a lot of work where you can learn the relationship between the force readings and the pose misalignment. So, proposing a LLM solution where the LLM can specify some ad-hoc rules for impedance and help in a search-based strategy is not a good policy for contact-rich tasks.
> So, in the rebuttal your claim that your proposed method can do force-reading based contact-rich tasks is not correct. Rather you can only tune the impedance parameters for some class of tasks (these are basically trivial contact-rich tasks).

---

### Author Response · Authors · 2023-11-23

We greatly appreciate the valuable input from all reviewers. Your suggestions have undoubtedly enhanced our submission. In response to your feedback, we have added the following:

- A new experiment where we generalize the prompt examples used in the cable routing and unrolling task to
- Example with spatial prompts (https://dex-code-gen.github.io/dex-code-gen/)
- Changes in presentation that are highlighted in red in our revised draft

We hope our revisions meet your concerns. In the words of the reviewers, we present a "thoughtful design of the action space (conditional compliant moves) to enable successful completion of contact-rich manipulation tasks by policies generated by LLMs." (7Rys) This is the first paper to study the generation of robot policy code using an LLM for contact-rich manipulation tasks and, surprisingly, we find that it is possible for LLMs to generate useful policy code by composing examples of conditional compliant primitives.

---

### Meta-Review · Area_Chair_AYFE · 2023-12-03

**Metareview:**

The paper received ratings indicating a tendency towards rejection (5,5,5,1). The reviewers had several concerns such as:
- The LLM-based method not being general enough,
- No handling of partial observability,
- No validation of hint inputs,
- Unfair comparisons with the LLM-based approach,
- Considering perception as a solved problem.

The AC reviewed the paper, the reviews, and the rebuttal. The rebuttal is not convincing since some of the issues are fundamental issues that cannot be rectified through the rebuttal, and they should be addressed before publication. The paper requires a major revision. Hence, rejection is recommended.

**Justification For Why Not Higher Score:**

The paper has various issues (mentioned above), and those issues should be addressed to meet the bar for acceptance.

**Justification For Why Not Lower Score:**

N/A

---

### Decision · Program_Chairs · 2024-01-16

Reject